# Self-supervised Out-of-distribution Detection for Cardiac CMR Segmentation

**Camila Gonzalez**  camila.gonzalez@gris.informatik.tu-darmstadt.de  and
**Anirban Mukhopadhyay**  anirban.mukhopadhyay@gris.informatik.tu-darmstadt.de
*Technical University of Darmstadt, Karolinenpl. 5, 64289 Darmstadt, Germany*

## Abstract

The segmentation of cardiac structures in Cine Magnetic Resonance imaging (CMR) plays an important role in monitoring ventricular function, and many deep learning solutions have been introduced that successfully automate this task. Yet due to variabilities in the CMR acquisition process, images from different centers or acquisition protocols differ considerably. This causes deep learning models to *fail silently*. It is therefore crucial to identify out-of-distribution (OOD) samples for which the trained model is unsuitable. For models with a self-supervised proxy task, we propose a simple method to identify OOD samples that does not require adapting the model architecture or access to a separate OOD dataset during training. As the performance of self-supervised tasks can be assessed without ground truth information, it indicates during test time when a sample differs from the training distribution. The proposed method combines a voxel-wise uncertainty estimate with the self-supervision information. Our approach is validated across three CMR datasets and two different proxy tasks. We find that it is more effective at detecting OOD samples than state-of-the-art post-hoc OOD detection and uncertainty estimation approaches.

**Keywords:** out-of-distribution detection, self-supervision, distribution shift

## 1. Introduction

Despite significant advances in diagnostic deep learning research, the adoption of learning-based systems in clinical practice is very limited. One reason for this is the inability of models to generalize to out-of-distribution (OOD) samples in real clinical settings, coupled with their tendency to produce overconfident predictions. Most deep learning systems are evaluated on test data similar in distribution to that used for training. When testing takes place on data gathered from different pieces of equipment or with a different protocol, there is a noticeable drop in performance (Glocker et al., 2019).

Cardiac Cine Magnetic Resonance imaging (CMR), the gold-standard for non-invasive volumetric quantification, is particularly prone to shifts in image properties. The acquisition process requires breath-holding, which is difficult for patients with arrythmias. As a consequence, variations in image quality are magnified (Oksuz et al., 2019; Ruijsink et al., 2020). Automatic cardiac segmentation that generalizes well to unseen manufacturers is still an open challenge (Bevandić et al., 2019; Yan et al., 2020). Clinical deployment of deep neural networks (DNNs) would comprise a two-step process where the plausibility of a model output being correct is considered alongside the prediction. Observing softmax outputs is not sufficient, as DNNs produce overconfident predictions for OOD data (Hein et al., 2019). Fig. 1 shows how the segmentation performance of a U-Net deteriorates silently on

OOD data. As OOD detection is a secondary goal, an ideal detector would integrate into any existing model and require no modifications in the architecture or training procedure.

In this work, we explore how self-supervision can help uncover OOD samples for the task of left ventricular blood pool segmentation, which is often utilized clinically to calculate parameters such as Ejection Fraction. DNNs only produce meaningful outputs for in-distribution (ID) data (Su et al., 2020). This manifests in a drop in performance for OOD samples and, accordingly, a higher loss between the predicted and target values. While the loss cannot be calculated during inference for supervised tasks, it *can* be for self-supervised tasks that derive target values from the input images. For self-supervised models, this opens the possibility to leverage the test-time performance as a signal for the identification of OOD samples without needing any manual annotations or OOD training data.

Our proposed method uses the value of the self-supervision loss in combination with post-hoc uncertainty estimation. While other works have used the self-supervision loss to detect OOD samples in classification tasks, we adopt this idea for medical image segmentation. Unlike current state-of-the-art, the proposed approach does not require a specific proxy task, or training the model with the explicit goal of OOD detection, and is therefore applicable to a wide array of self-supervised architectures. The proposed method outperforms state-of-the-art post-hoc approaches for OOD detection and uncertainty estimation across three CMR datasets and for two different proxy tasks: edge detection and contrastive learning. Our main contributions are: (A) the introduction of self-supervision as a lightweight OOD detector for cardiac CMR segmentation and (B) a thorough evaluation of OOD detection methods on CMR imaging for three datasets and two different self-supervised architectures.

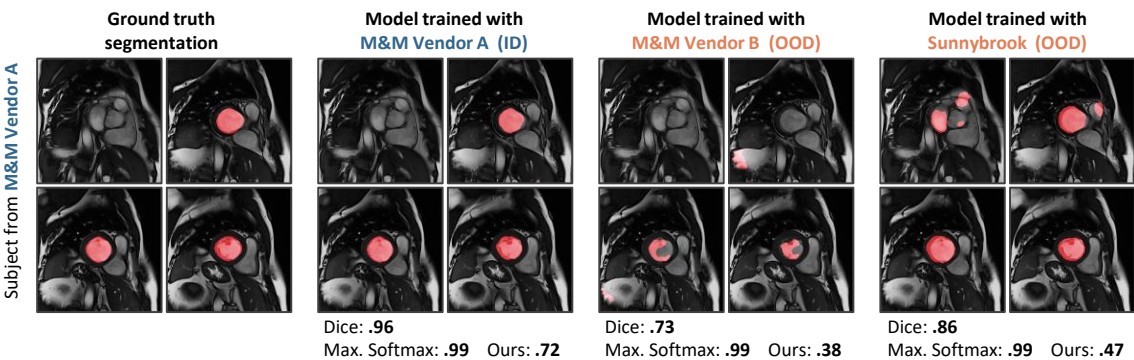

Figure 1: Distribution shift causes a deterioration on the left ventricular blood pool segmentation for a subject from the *Multi-Centre, Multi-Vendor and Multi-Disease (M&M) Vendor A* dataset, but traditional confidence quantification fails silently.

## 2. Related Work

In this section, we review relevant related work for self-supervision and OOD detection.

**Self-supervision** methods combine the training for the regular *target task* with a *proxy task*. Whereas the target task is usually supervised, the proxy task does not require manual

annotations, i.e. the target value can be derived from the input. For the sake of brevity we refer to Asano et al. (2019) and Zhang et al. (2019) for a detailed description of self-supervision in image segmentation.

In the field of **out-of-distribution detection**, several methods look at network outputs to detect novel samples. Hendrycks and Gimpel (2016) introduce the baseline of using the distribution of softmax values as an indicator for novelty. Guo et al. (2017) find temperature scaling to be an effective DNN calibration method. Liang et al. (2018) introduce the *ODIN* method, which extends temperature scaling by adding small adversarial-like perturbations to the inputs during inference which increase the separation between ID and OOD softmax values. Lee et al. (2018b) use the class-conditional distribution of neural activations to detect OOD samples. Other methods – that do not work in a post-hoc basis – use OOD data during training to explicitly train an outlier detector (Hendrycks et al., 2018; Lee et al., 2018a; Mohseni et al., 2020; Vyas et al., 2018; Bevandić et al., 2019). Related to the task of OOD detection is **uncertainty estimation**. Popular methods include Monte Carlo (MC) Dropout (Gal and Ghahramani, 2016) and Deep Ensembles (Lakshminarayanan et al., 2017). Several publications look at their effectiveness in the field of medical image segmentation, and find that ensembles are most reliable, though MC Dropout is also effective (Jungo and Reyes, 2019; Jungo et al., 2020; Mehrtash et al., 2020). Other methods have shown better performance in some cases, but require special training considerations (Blundell et al., 2015; Kohl et al., 2018; Monteiro et al., 2020).

Some research delves into **OOD detection in self-supervised models**. Pidhorskyi et al. (2018) use the reconstruction error of an autoencoder to assess novelty. Winkens et al. (2020) and Wu and Goodman (2020) augment classification networks with a contrastive learning term and estimate the density on different feature spaces. Similar to us, Golan and El-Yaniv (2018) train a multi-head model, where one head performs image classification and the second learns to detect image transformations, and calculate the novelty through the softmax outputs. Hendrycks et al. (2019) improve OOD detection by training a classifier with a proxy rotation estimation loss. For image segmentation, Xia et al. (2020) calculate the reconstruction error between the original image and a synthesized version.

Unlike other approaches, our proposed method does not require the use of a particular proxy task, and works entirely in a post-hoc manner. This ensures the applicability to a variety of deployed learning systems that include a self-supervised component. In terms of application we focus on semantic segmentation, and evaluate our method on datasets which solve the same semantic task (left ventricular blood pool segmentation) but differ in terms of acquisition vendor and center. Our research is, to our knowledge, the first to utilize self-supervision losses for OOD detection in medical image segmentation.

## 3. Methods

Consider a model $\mathcal{F}$ trained with $n$ samples $\{x_i\}_{i=1}^n$. The goal of OOD detection is to identify – during deployment – new samples that variate significantly from the training distribution. For this, a continuous *novelty* function $\mathcal{N} : \mathcal{X} \rightarrow \mathbb{R}$ and a threshold $\psi$ are defined so that $x_i$ is classified as out-of-distribution if $\mathcal{N}(x_i) \geq \psi$. The expectation is that real-world OOD samples are flagged for which the model produces unreliable predictions. In this section, we describe our proposed method to detect OOD samples in a post-hoc

manner for models trained with a self-supervised proxy task. We start by introducing the two architectures we explore in this work, and then explain the process of OOD detection.

### 3.1. Self-supervised Learning

A task is said to be *self-supervised* if the target information is generated by the learning system. Increasingly, DNNs for semantic segmentation are being augmented with self-supervision (Wang et al., 2020; Pan et al., 2020) in order to leverage non-annotated data or shape the feature space. In this work, we explore **edge detection** and **contrastive learning**. These proxy tasks are well-suited to the segmentation of cardiac structures as they encourage learning geometrically-aware features that disregard image quality information (Chu et al., 2020; Winkens et al., 2020; Sahu et al., 2020). However, the novelty metric we introduce in Sec. 3.2 can be calculated for models trained with any self-supervised task.

**Contrastive learning** teaches the model to distinguish between different data points in the training set, while at the same time learning a semantically meaningful feature space that disregards certain transformations. Inspired by Winkens et al. (2020), we transform an original image $x_i$ into $\mathcal{T}(x_i) = \overline{x_i}$. During training, we maximize the cosine similarity between $x_i$ and $\overline{x_i}$ in the feature space and minimize the similarity between $x_i$ and a second image $x_j$. For function $\mathcal{T}$, we use implementations from the *TorchIO* library (version 0.17.46) (Pérez-García et al., 2020). We randomly apply *RescaleIntensity*, *RandomGamma*, *RandomMotion*, *RandomBiasField*, *RandomNoise* and *RandomBlur* operations, each with a probability of $p = 0.5$. Features $z_i$ are extracted from the output of the encoder $\mathcal{E}$. Eq. 1 defines the contrastive loss $\mathcal{L}_{ss}^{C}$, and the architecture is displayed in Fig. 2 (left).

$$\mathcal{L}_{ss}^{C}(x_i, x_j) = \mathcal{L}_{sim}(\mathcal{E}(x_i), \mathcal{E}(x_j)) - \mathcal{L}_{sim}(\mathcal{E}(x_i), \mathcal{E}(\mathcal{T}(x_i))), \quad \mathcal{L}_{sim}(z_i, z_j) = \frac{z_i \cdot z_j}{\|z_i\|_2 \cdot \|z_j\|_2} \quad (1)$$

The goal of **edge detection** is to extract a mask of edges $\widehat{h}_i$ from image $x_i$. We train a standard two-headed architecture consisting of a shared encoder $\mathcal{E}$ and two decoders, $\mathcal{G}$ for the segmentation task and $\mathcal{H}$ for edge detection. Fig. 2 (right) outlines the proposed architecture. We train both heads with a combined loss of Dice ($\mathcal{L}_{Dice}$) and binary cross entropy ($\mathcal{L}_{BCE}$) weighted equally. To produce target masks $h_i$ in a deterministic manner, we use the *Canny Edge* detector (Canny, 1986) of the *Scikit Learn* (Pedregosa et al., 2012) library (version 0.24.1) with lower and upper bounds of, respectively, 150 and 200. During inference, we treat the edge detection loss $\mathcal{L}_{ss}^{E}$ (Eq. 2) as a component of our novelty metric.

$$\mathcal{L}_{ss}^{E}(x_i, h_i) = \mathcal{L}_{Dice}(\mathcal{H}(x_i), h_i) + \mathcal{L}_{BCE}(\mathcal{H}(x_i), h_i) \quad (2)$$

### 3.2. Novelty Estimation

For detecting OOD samples during inference we combine uncertainty estimates with the loss of the self-supervised proxy task. Uncertainty estimation produces good calibrations in ID data, but often fails in the presence of dataset shift (Ovadia et al., 2019). We expect dataset shift to manifest in an unusually large self-supervision loss (Su et al., 2020) that compensates for the decreased ability to detect uncertain cases of uncertainty estimation methods. By combining these two factors, we obtain a reliable detection signal.

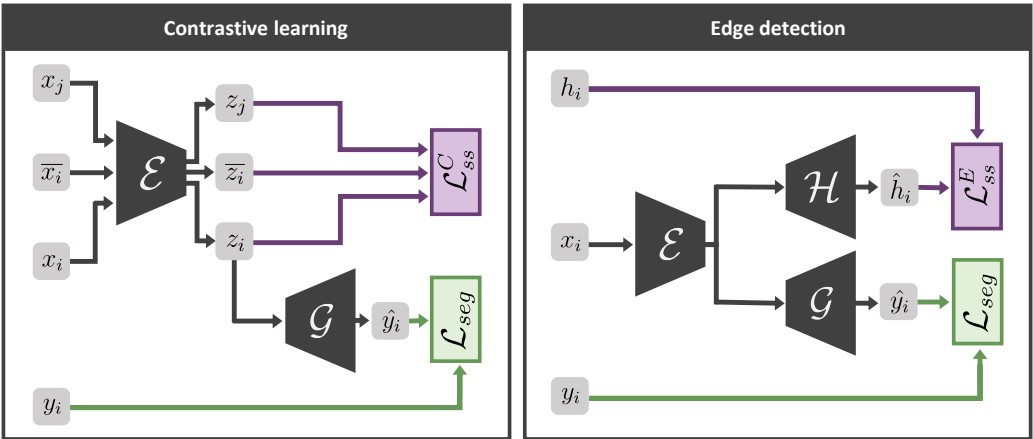

Figure 2: Two self-supervision architectures are explored in this work. Left: features are extracted for $x_i$, $\mathcal{T}(x_i) = \overline{x_i}$ and $x_j$ to calculate a contrastive loss term. Right: network with an additional decoder head for the task of edge detection.

As we aim to find a flexible post-hoc method applicable to most learning-based systems, we explore two different types of uncertainty estimation. **MC Dropout** (Gal and Ghahramani, 2016) involves performing several forward passes with dropout during test time. The method can be applied to any model that uses dropout layers, which includes most modern architectures. **Deep Ensembles** – the practice of training several networks and averaging their predictions – have consistently shown the best performance in uncertainty estimation (Jungo et al., 2020; Mehrtash et al., 2020). They are also a straightforward way to improve prediction performance and therefore often used in practice. In the event that several trained models are present, we propose using this method as an uncertainty estimate.

During inference, the novelty of a test subject is assessed by combining the self-supervised loss $\mathcal{L}_{ss}$ with uncertainty estimation. The $\mathcal{L}_{ss}$ loss is calculated in the same way as during training. For the experiments performed in this work, either $\mathcal{L}_{ss}^C(x_i, x_j)$ or $\mathcal{L}_{ss}^E(x_i, h_i)$ are calculated depending on the model architecture. In the first case, we use a different subject from the same dataset as $x_j$. For 2D models, the loss for a test subject is the average across slices, as is also the case during training. As the uncertainty estimation component we take the voxel-wise standard deviation between model predictions, which is averaged over all voxels to produce a subject-level score. Different predictions are obtained by performing MC Dropout or, if ensembles are available, by making a prediction with each model. We define the proposed novelty function $\mathcal{N}$ in Eq. 3, where $K$ is the number of trained models or dropout forward passes and $N$ is the number of voxels $x_{i,j}$ in an image $x_i$.

$$\mathcal{N}(x_i) = \lambda \mathcal{L}_{ss}(\cdot) + \frac{1}{N} \sum_{j=1}^{N} \sqrt{\frac{1}{K} \sum_{k=1}^{K} \left( x_{i,j}^k - \mu_{i,j} \right)^2}, \quad \mu_{i,j} = \frac{1}{K} \sum_{k=1}^{K} x_{i,j}^k \qquad (3)$$

## 4. Experimental Setup and Results

We use three CMR datasets. The first two are part of the *Multi-Centre, Multi-Vendor and Multi-Disease Cardiac Segmentation* (M&M) dataset (Campello and Lekadir, 2020) and contain healthy subjects as well as subjects with hypertrophic and dilated cardiomyopathies. We use the data for vendors $A$ and $B$, for which ground truth segmentations are available. The images were acquired with *Siemens Avanto* and *Philips Achieva* scanners, respectively, at different centers. Each dataset contains 75 subjects. Lastly, we use the *Sunnybrook Cardiac Data* (Radau et al., 2009), acquired at a different center with a *General Electric Signa* scanner. The data consists of 45 scans from healthy as well as diseased subjects suffering from hypertrophy and heart failure. All images were acquired with 1.5T fields strength. We extract from each subject the segmented diastolic and systolic phase volumes.

We train a slice-by-slice U-Net with five encoding blocks based on the implementation by Pérez-García (2020). Images are center-cropped to $256 \times 256$. Each model is trained for 200 epochs with the *PyTorch Adam* optimizer. For the edge detection task, the encoder is shared and the decoder is replicated from the point with minimum spatial resolution. Refer to Appendix A for an overview of segmentation performance in ID and OOD data. Note that the results on the target task change slightly due to the incorporation of self-supervision.

We compare the proposed method against taking the inverse maximum softmax value (Hendrycks and Gimpel, 2016) (reported as **Max. Softmax**), temperature scaling (**Temp. Scaling**) (Guo et al., 2017) and the **ODIN** method (Liang et al., 2018); as well as against the corresponding uncertainty estimation (**MC Dropout** and **Ensemble**) and using only the self-supervised loss as a novelty estimate (**SS Loss**). When necessary, we average voxel-wise estimates to produce a volume-wise novelty score. We refer to our method variations using and not using ensembles as **Ours E** and **Ours**, respectively. We further specify in parenthesis whether the model learned a contrastive (C) or edge detection (E) task.

In turn, we consider each of the three datasets as ID and the other two as OOD. We divide the ID cases into three folds to perform cross-validation. For each cross-validation run, we train a model with the *ID train data* made out of two folds and evaluate it with the third fold, which is the *ID test data*. For OOD detection, we use one OOD dataset and the ID train data to select the best hyperparameters and evaluate the detection performance on the second OOD dataset and the ID test samples. We average the results of using each of the two OOD datasets for the evaluation, and report the mean and standard deviation of the three-fold cross-validation. Refer to Appendix D for a graphical illustration of our evaluation strategy. The following hyperparameters are tested: $T \in \{1e1, 1e2, 1e3\}$ for temperature, $\varepsilon \in \{1e-1, 1e-2, 1e-3\}$ for perturbation magnitude (ODIN), $p \in \{0.3, 0.5, 0.7\}$ for dropout probabilities and $\lambda \in \{1e0, 1e2, 1e4\}$ for weighting magnitudes.

We train ensembles with $K = 3$ models and perform $K = 30$ MC Dropout passes. We select the threshold $\psi$ that achieves a 95% True Positive Rate (TPR) in the in-distribution train data, and flag samples as OOD when $\mathcal{N}(x) \geq \psi$. Reported are the Detection Error as defined by Liang et al. (2018) and the False Positive Rate (FPR) at 95% TPR.

### 4.1. Results for Contrastive Learning Models

We start by analyzing the results of OOD detection methods for the models trained with a contrastive learning loss component. Table 1 summarizes our findings. We see that for

Table 1: OOD Detection Error and FPR at 95% TPR for models trained with a contrastive learning loss term (lower is better). The mean and standard deviation are reported of testing with each OOD dataset and performing three-fold cross validation.

| Method | M&M Vendor A | | M&M Vendor B | | Sunnybrook | |
|---|---|---|---|---|---|---|
| | Error | FPR | Error | FPR | Error | FPR |
| **Max. Softmax** | .48 ±.00 | .93 ±.01 | .51 ±.02 | .90 ±.02 | .53 ±.00 | .91 ±.09 |
| **Temp. Scaling** | .51 ±.01 | .93 ±.01 | .51 ±.02 | .93 ±.01 | .47 ±.01 | .90 ±.02 |
| **ODIN** | .43 ±.02 | .84 ±.03 | .49 ±.00 | .87 ±.01 | .51 ±.01 | .87 ±.02 |
| **SS Loss (C)** | **.33 ±.03** | .61 ±.04 | .36 ±.11 | .60 ±.17 | .50 ±.04 | .91 ±.02 |
| **MC Dropout** | .45 ±.01 | .85 ±.05 | .38 ±.10 | .72 ±.20 | .21 ±.02 | .23 ±.09 |
| **Ours (C)** | **.33 ±.03** | **.60 ±.05** | **.33 ±.12** | **.58 ±.18** | **.19 ±.02** | **.19 ±.09** |
| **Ensemble** | .46 ±.02 | .86 ±.01 | .44 ±.03 | .37 ±.08 | **.26 ±.01** | .06 ±.02 |
| **Ours E (C)** | **.32 ±.05** | **.49 ±.13** | **.26 ±.05** | **.17 ±.04** | .28 ±.01 | **.05 ±.00** |

all datasets, the popular temperature scaling and ODIN methods perform poorly. This may be due to the fact that both methods are developed for the classification task and not segmentation, where different voxels may be more or less significant for determining whether a sample is in-distribution. Our proposed method results in a lower detection error and FPR than all baselines both in cases where ensembles are available and when they are not. Only in dataset *Sunnybrook* does the ensemble alone achieve a lower detection error than the proposed method. As expected, considering the deviation between ensembles as an uncertainty estimation component leads to better results than applying MC Dropout. However, this method variation is only applicable if multiple models have been trained.

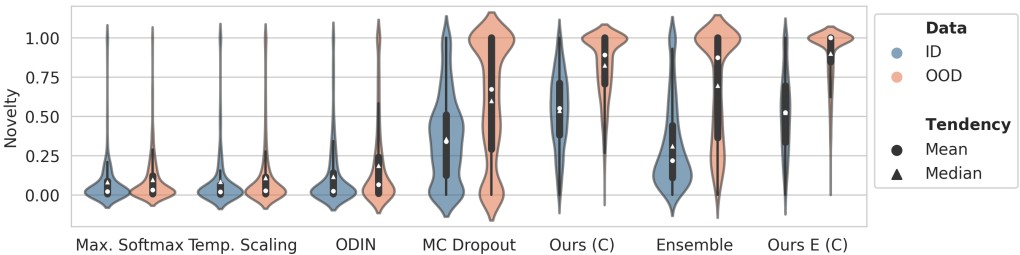

Figure 3: Distribution of novelty scores for contrastive learning models (lesser overlap is better). The scores for ID and OOD data are aggregated for all experiments and normalized to $[0, 1]$ by taking the range of the ID training set.

Fig. 3 illustrates the ranges that different novelty scores occupy, normalized by taking the minimum and maximum novelty for ID train data, so that different methods are comparable. Ideally, novelty scores would cluster close to one (upper plot segment) for OOD data, and there would be a minimal overlap between ID and OOD scores. By observing the boxes

ranging from the first to the third quantiles we notice that the proposed method achieves the best separation between ID and OOD novelty scores in its two variations.

### 4.2. Results for Architectures with Edge Detection

Table 2 compiles the results for models trained with an edge detection proxy task. Despite this being a very different task and self-supervision loss, the proposed method still performs best in all but one cases. However, the method shows its limitations for models trained with data from *M&M Vendor B*. This indicates that although our method is suited to any self-supervised task, some tasks may be more helpful than others.

Table 2: OOD Detection Error and FPR at 95% TPR ($\pm$ standard deviation) for models trained with an edge-detection proxy task (lower is better).

| Method | M&M Vendor A | | M&M Vendor B | | Sunnybrook | |
|---|---|---|---|---|---|---|
| | **Error** | **FPR** | **Error** | **FPR** | **Error** | **FPR** |
| **Max. Softmax** | .49 $\pm$.00 | .97 $\pm$.00 | .49 $\pm$.01 | .95 $\pm$.04 | .50 $\pm$.00 | .96 $\pm$.01 |
| **Temp. Scaling** | .51 $\pm$.01 | .87 $\pm$.02 | .51 $\pm$.03 | .91 $\pm$.01 | .48 $\pm$.02 | .92 $\pm$.04 |
| **ODIN** | .47 $\pm$.02 | .90 $\pm$.01 | .48 $\pm$.03 | .89 $\pm$.02 | .48 $\pm$.01 | .92 $\pm$.00 |
| **SS Loss (E)** | .33 $\pm$.01 | .66 $\pm$.01 | .55 $\pm$.03 | .99 $\pm$.01 | .29 $\pm$.01 | .53 $\pm$.04 |
| **MC Dropout** | .43 $\pm$.04 | .81 $\pm$.04 | **.44 $\pm$.06** | **.33 $\pm$.33** | **.28 $\pm$.03** | .28 $\pm$.16 |
| **Ours (E)** | **.32 $\pm$.01** | **.63 $\pm$.01** | **.44 $\pm$.05** | .81 $\pm$.16 | **.28 $\pm$.02** | **.25 $\pm$.14** |
| **Ensemble** | .39 $\pm$.04 | .68 $\pm$.14 | **.45 $\pm$.03** | .45 $\pm$.02 | .37 $\pm$.13 | .51 $\pm$.49 |
| **Ours E (E)** | **.32 $\pm$.01** | **.55 $\pm$.08** | **.45 $\pm$.03** | **.44 $\pm$.02** | **.25 $\pm$.01** | **.23 $\pm$.22** |

## 5. Conclusion

Automatic segmentation of cardiac structures in CMR data could significantly alleviate the burden of clinicians. Competitive performance has been achieved by DNNs, but as long as these are susceptible to domain shift their applicability is limited. One way to approach this is by identifying OOD samples during deployment. For self-supervised models, combining the test-time value of the proxy loss with uncertainty estimation forms a reliable and lightweight novelty score. This finding is significant when considering the surge in popularity of self-supervision and introduces a further benefit of including a proxy term in DNN training. The proposed method can augment a wide array of learning-based systems, although for fully-supervised models incorporating a proxy task can have unintended effects in the target task. Future work should contemplate whether our results extend to other proxy tasks and anatomies. As it requires minimal overhead, we hope that monitoring the proxy loss during deployment becomes a widespread method for quality assurance.

### Acknowledgments

This work was supported by the Bundesministerium für Gesundheit (BMG) with grant [ZMVI1-2520DAT03A].

## Appendix A. Segmentation Performance of Trained Models

Table 3 showcases the Dice coefficient for left ventricular blood pool segmentation for models trained with two proxy tasks (contrastive and edge detection), as well as without any proxy task. In the diagonal, the results are displayed of testing each model with ID data.

Table 3: Mean Dice for models trained with a contrastive learning loss component (first row), edge detection (second row) and no self-supervised loss (third row). Reported are the mean and standard deviation of three cross-validation runs.

|  | Data | $\mathcal{F}$ trained with M&M Vendor A | $\mathcal{F}$ trained with M&M Vendor B | $\mathcal{F}$ trained with Sunnybrook |
|---|---|---|---|---|
| $\mathcal{L}_{ss}^{C}$ | M&M Vendor A | **.85 ±.02** | .37 ±.05 | .57 ±.02 |
|  | M&M Vendor B | .71 ±.01 | **.87 ±.02** | .44 ±.10 |
|  | Sunnybrook | .57 ±.03 | .14 ±.04 | **.83 ±.02** |
| $\mathcal{L}_{ss}^{E}$ | M&M Vendor A | **.83 ±.04** | .36 ±.05 | .50 ±.05 |
|  | M&M Vendor B | .65 ±.02 | **.86 ±.02** | .35 ±.15 |
|  | Sunnybrook | .60 ±.03 | .09 ±.03 | **.82 ±.01** |
| No $\mathcal{L}_{ss}$ | M&M Vendor A | **.86 ±.02** | .42 ±.05 | .60 ±.06 |
|  | M&M Vendor B | .71 ±.07 | **.87 ±.06** | .36 ±.08 |
|  | Sunnybrook | .53 ±.02 | .16 ±.08 | **.80 ±.06** |

## Appendix B. Novelty Distribution for Edge Detection Models

Fig. 4 displays the distribution of novelty scores for models with an edge detection proxy task. We see that the amount of overlap between ID and OOD data is more pronounced than for contrastive learning models (see Fig. 3). The variant of our method that uses ensembles (*Ours E (E)*) is the only approach that achieves a good separation.

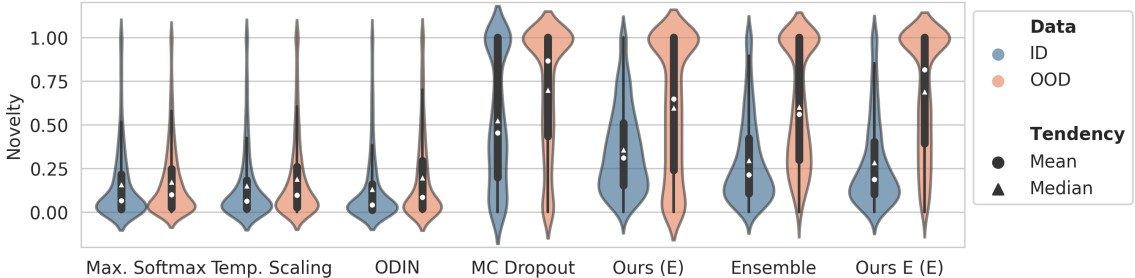

Figure 4: Distribution of novelty scores for models with an edge detection proxy task (lesser overlap is better). The novelty scores for ID and OOD data are aggregated for all experiments. The scores were normalized to $[0, 1]$.

## Appendix C. Generation of Target Data for Proxy Tasks

Fig. 5 displays exemplary data generated to train the proxy tasks explored in this work. The first column showcases slices from the *M&M Vendor B* dataset with overlayed ventricle blood pool segmentation (in red). The second column shows the same slices but with overlayed edge masks. Finally, the third column illustrates possible results of applying the transformation $\mathcal{T}(x_i) = \overline{x_i}$.



Input image slices $x_i$     Edge masks $h_i$     Transformed image slices $\overline{x_i}$



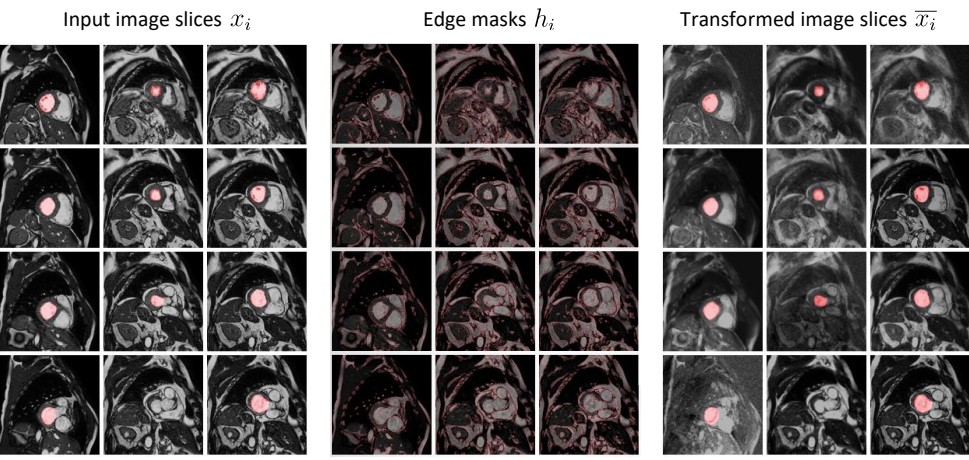

Figure 5: From left to right: image $x_i$ with overlayed left ventricle blood pool segmentation ($y_i$), $x_i$ with overlayed edges $h_i$ and transformed image $\overline{x_i}$ with overlayed $y_i$.

## Appendix D. Evaluation Strategy

Fig. 6 graphically illustrates our evaluation setup with three datasets for one cross-validation run. In turn, each dataset is considered ID and is divided into *ID train* and *ID test* data. The ID train data is used to train the model, as well as to set hyperparameters alongside one OOD dataset. The detection performance is reported in the ID test data and the second OOD dataset. The results of using each OOD dataset for each purpose are averaged.

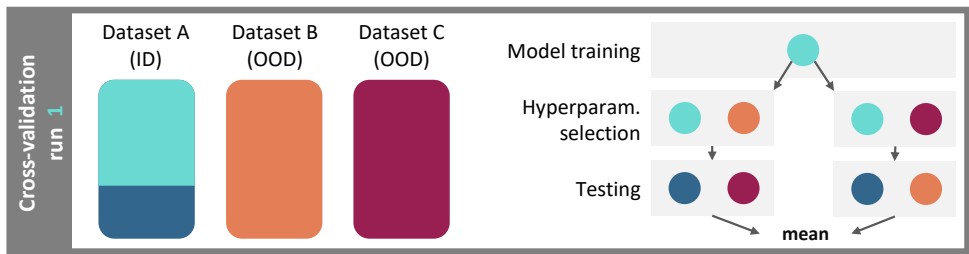

Figure 6: Graphical illustration of the evaluation strategy.

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
