# OpenReview forum: "Self-supervised Out-of-distribution Detection for Cardiac CMR Segmentation"
_MIDL.io/2021/Conference — MIDL 2021_

### Official Review · AnonReviewer2 · 2021-02-28

**Confidence:** 5
**Preliminary Rating:** 2

**Summary:**

The paper presents two approaches to OoD detection involving training with a self supervised task which can then be used to detect OoD samples at test time. The focus is on segmentation.

Their stated contributions are:

"The introduction of self-supervision as a lightweight OOD detector for cardiac CMR segmentation."
"A thorough evaluation of OOD detection methods on CMR imaging for three datasets and two different self-supervised architectures."


**Strengths:**

Interesting use of an edge detector as a proxy task to detect outliers.
Methods evaluated on multiple external datasets.

..............................................................................

**Weaknesses:**

Presentation unclear.
The first time that the proposed method is stated in details is page 4.
Not clear how OoD is defined in the experiments.

..............................................................................

**Deanonymize Review:**

no

**Detailed Comments:**

It is difficult to under the core self-supervised part which is a primary contribution of the work. As I understand it the proxy tasks to score the OoD samples are a contrastive loss and predicting known edges. Please make this clear in the introduction so it is easier for the reader to get the core of the paper quickly.

It is not clear how the contrastive loss is used to score outlier samples. Maybe defining a clear score function would make this clear. Also, the transformation functions seem important and should be part of the main text. Possibly moving the related work to the appendix is a solution for space.

For edge detection it is more clear but it would be better to explicitly define the scoring function.

I don't see what OoD samples are used. The paper describes in-distribution datasets but does not describe your OoD validation or test data as far as I can see.

For Fig3 it would be easier to interpret if you included p-values. Otherwise is it not so clear how to interpret the results.

I would look into this work: https://arxiv.org/abs/1809.04729 as a framework for evaluating OoD methods.

Also, having more baseline methods such as a binary classifier trained on the latent space of the model would provide a more compelling evaluation.


**Justification Of The Preliminary Rating:**

Presentation is not clear.
Method is not fully explained.
Evaluation is unclear.
Conclusions are not strong enough.

...................................................................................

**Paper Type:**

methodological development

**Questions To Address In The Rebuttal:**

Make the introduction more direct to clearly explain what the paper is proposing in detail.
Make stronger and more direct conclusions about what the take away from this paper is.
Clearly describe the OoD datasets used and how they relate to the deployment situations that you aim to address.
Clearly define how OoD samples are scored during test time.


**Special Issue:**

no

---

> ### Author Response · Authors · 2021-03-16
> **Response to AnonReviewer2 (1/2)**
>
> We thank the reviewer for pointing out the lack of clarity in the original manuscript. We agree that the manuscript was ambiguous in several critical points and did not state the methodological contributions and conclusions in a sufficient manner. We hope that we have addressed these concerns in the revised manuscript and the comments below.
>
> **Regarding the late introduction of the method**
>
> We agree with the reviewer that in the original manuscript, the method was not sufficiently described in the introduction. We have rephrased the last two paragraphs of the introduction (page 2) to state the motivation and workings of the proposed method more clearly.
>
> **Regarding the conclusion and main takeaway from the paper**
>
> We thank the reviewer for their feedback. We agree that our original conclusion did not clearly state our main contributions and takeaways. We have adapted our conclusion (page 8) and hope that the contributions of the paper are now more clear.
>
> **Regarding the evaluation strategy**
>
> As several reviewers have also pointed out, the description of our evaluation strategy was not clearly stated in the original manuscript. Indeed, our evaluation setup is very similar to that proposed by Shafaei et al. (2019). Inspired by the graphical illustration of the paper, we now illustrate our evaluation setup in a similar fashion in Appendix D. We have also rephrased our description on page 6.
>
> While we use a similar experimental setup as described in the suggested publication (Shafaei et al., 2019), neither ours nor any of the compared methods require OOD data during training. We therefore use one OOD dataset and the ID train data to set hyperparameters, and evaluate with the ID test data and the second OOD dataset. In Shafaei et al. (2019), the hyperparameters are instead selected by following best practices as described in previous publications.
>
> **Regarding the used datasets**
>
> Indeed, it was not clear in the original manuscript that we consider each dataset to be ID or OOD in turn. We now explain this more clearly in the “Experimental Setup and Results” section (page 6). In Tables 1 and 2, the dataset mentioned in the column heading is always considered ID, and the others are OOD. Regarding the used datasets, we have included further details in page 6 (top).
>
> **Regarding the scoring of test samples**
>
> Based on the feedback, we have added a paragraph at the end of the methodology section (page 5) to clarify how the novelty is calculated for a test sample. Samples are scored during inference according to the novelty function defined in Equation 3. The novelty function contains two components: the self-supervised loss and the voxel-wise uncertainty. For both contrastive learning and edge detection, the loss function that was used during training (Equations 1 and 2, respectively) is calculated for the test subject. The uncertainty estimate, averaged over all voxels, makes up the second component of the novelty function.
>
> **Regarding the description of the the proxy tasks in the introduction**
>
> Indeed, this was missing from the introduction in the original manuscript, where we only mentioned that we evaluated our method with two self-supervised architectures. We have corrected our introduction to state that we evaluate our method on edge detection and contrastive learning proxy tasks. However, we would like to emphasize that we do not propose using the specific proxy tasks explored in this work. Increasingly, segmentation methods include a self-supervised component for various reasons. The main takeaway from the paper is the way in which a self-supervised loss can be used at test time for OOD detection. Segmentation models that already contain a self-supervised component can thus be augmented with an OOD detector with minimal overhead.

---

> ### Author Response · Authors · 2021-03-16
> **Response to AnonReviewer2 (2/2)**
>
> **Regarding the interpretation of Figure 3**
>
> We thank the reviewer for the feedback. We agree that the intent of the original plots was not sufficiently stated. We have replaced the boxplots with violin plots and elaborate on the main conclusions in the text in the hope to make this clearer. What we aim to show in the plots is that our proposed method assigns OOD data much higher novelty values than ID data, and that it displays the least overlap between ID and OOD novelty ranges from all observed methods.
>
> **Regarding the selected baseline methods**
>
> We agree with the reviewer that a more thorough comparison would be beneficial. Since our proposed method does not require OOD data to train, we did not compare our performance with a binary classifier. But we must emphasize that we compare the method against six approaches, as presented in the manuscript. Since all these methods output a continual score that can be interpreted as a novelty metric, comparing with a binary classifier would require a different evaluation setup. The six baselines we include, as well as our method, can be calculated in a purely post-hoc basis without requiring OOD data and output continual novelty values.

---

### Official Review · AnonReviewer4 · 2021-03-01

**Confidence:** 4
**Preliminary Rating:** 3
**Recommendation:** Poster
**Final Rating:** 4

**Summary:**

The paper at hand addresses the problem of OOD sample detection for deep learning-based CMR segmentation. The authors propose to use a combination of common uncertainty measures (MC dropout and ensembles) and self-supervised auxiliary tasks (edge detection and contrastive loss). The authors compare multiple variants of their approach and other methods in an extensive evaluation. The evaluation is performed on public datasets with data from different vendors. The method generally performs well and often better than other methods.

**Strengths:**

•	The paper is easy to read and interesting

•	The paper addresses the very relevant topic of OOD sample detection

•	The authors provide an extensive literature review

•	The author’s method description is detailed and mostly clear

•	The authors provide extensive results with comparisons to multiple other methods

•	The authors highlight both advantages and limitations of their methods


**Weaknesses:**

•	 The author’s description for data splitting and evaluation (4) is not entirely clear. Which dataset is used in which fold? Which datasets are used for hyperparameter tuning and which datasets are used for reporting results? The current description sounds like validation and test results are averaged, which the authors probably did not do

•	The authors should be moderate with their claim of novelty. The authors cite the paper Hendrycks 2019 which employs a self-supervised rotation loss which is also used for OOD scoring. I recommend to explicitly state that self-supervised losses have been used for OOD scoring and that the authors adopt this approach for CMR segmentation (see end of introduction).


**Deanonymize Review:**

no

**Detailed Comments:**

•	P.2: “Ejection Function” – the more common term is ejection fraction

**Final Rating Justification:**

The authors addressed the points I raised. Also, the authors provide adequate answers for the other reviewers' concerns.

**Justification Of The Preliminary Rating:**

The paper is interesting, methodologically mostly sound, and provides an extensive evaluation. I believe it is interesting for the MIDL community. My few concerns should be easy to address in the rebuttal.

**Paper Type:**

both

**Questions To Address In The Rebuttal:**

See the 2 weaknesses. From my point of view, both are easy to fix.

**Special Issue:**

no

---

> ### Author Response · Authors · 2021-03-16
> **Response to AnonReviewer4**
>
> We thank the reviewer for the thorough comments, as well as for raising important concerns about the description of the evaluation strategy and claim of novelty. We hope these are sufficiently addressed in the revised manuscript and the comments below.
>
> **Regarding the evaluation strategy**
>
> We thank the reviewer for the feedback. Indeed, the evaluation strategy was not sufficiently explained. We have rephrased the paragraph where we explain our evaluation setup in page 6, which was clearly ambiguous. We do not average our validation and test results but rather the test results using each of the two OOD datasets and the ID test data. We have also included a graphical illustration of our setup in Appendix D.
>
>
> **Regarding the claim of novelty**
>
> Indeed, as the reviewer accurately pointed out, other works such as Hendrycks (2019) have utilized self-supervision losses for OOD detection. While we state this at the end of the related work section in the original manuscript, it should definitely be mentioned alongside our statement of contribution earlier on. Hence, we have added this clarification at the end of the introduction (page 2).
>
>
> **Regarding detailed suggestions**
>
> We have corrected the use of the term *ejection fraction*. We thank the reviewer for pointing this out.

---

### Official Review · AnonReviewer1 · 2021-03-05

**Confidence:** 5
**Preliminary Rating:** 3
**Recommendation:** Oral
**Final Rating:** 4

**Summary:**

The paper addresses a well-known and important problem in medical image segmentation. Basically, the authors developed developed networks that benefits from self-supervision w.r.t. two different objectives and used them to assess the uncertainty associated with segmentation and detect out-of-distribution (OoD) examples via novelty scores. The application domain of the paper is Cardiac CMR segmentation. However, the ideas behind the proposed solution(s) have potential to be used in other specialties of medicine. Good work!

**Strengths:**

The paper is well-structured and well-written. This suggests its maturity.

Literature review is decent and points at the right direction.

Methods are clearly explained and easy to understand.

Experiments include 3 datasets with different manufacturers and sources.

Promising results with potential developments for other domains.

**Weaknesses:**

The segmentation networks here benefit from two auxiliary losses or self-supervision tasks. One is based on contrastive learning and the other on edge detection. My intuitive understanding is that edge detection provides a template for the segmentation branch to fill in. Contrastive task, however, makes it robust to certain types of transformations while reinforcing the representational capabilities of the network. Considering the overarching goal of the study, which is detecting OoD examples, contrastive task makes more sense to me. The other one also seems to work but maybe I like the contrastive one better. Anyways, the results show that the contrastive loss works preferably better too.

So, my question is along the lines of the contrastive learning. The contrastive learning algorithm typically require large minibatches or memory banks to exploit the similarities between a good amount of data points. These, in practice, can trick implementations. However, there is no information regarding these aspects in the current work.

Did you implement something new? Do you have an interesting design to share?

If you used an existing solution, what was that? Any modifications or special tricks played in this study?

I can see that you have space limitations here. But, 1-2 pages in Appendix could help.

In addition, training on images from Vendor A seems to allow for better generalization overall (I also checked Appendix B). What is special about Vendor A. What are the main differences from Vendor B. I am not asking you to give names but some comparison could help us understand what is going on better. For instance, old vs. new machine/technology, image resolution, technical capacities of operators, procedural differences, ...

3-fold cross validation: In general, DNNs exhibit a great deal of diversity in their function estimations due to various factors, such as optimisation trajectories, randomness in data (shuffling and augmentation), etc... Using a larger number of folds, e.g., 10, would allow for a better established results and improve the trust in findings here.

Finally, I need to ask an obvious question. Did you consider using both contrastive and edge detection tasks together? It seems like these could be combined into the same network architecture. These tasks could complement each other, possibly. Can you speculate a bit in this regard? What would happen if this was implemented? Did you have any concerns for not implementing it this way? I am just curious.

**Deanonymize Review:**

no

**Detailed Comments:**

Introduction, page 2
"Self-supervised tasks do not require manual annotations, and so the performance in test samples can be assessed."
This is sentence is unclear. So is the rest of paragraph. It makes a bit more sense after reading the manuscript. But, still a poor paragraph.

Fig. 1 caption: "U-Nets trained with OOD data" What? Then, OoD is in-distribution, right? Also, mention what M&M means in the caption. I had to search for it.

Page 3: "These increase maximum softmax probabilities to a greater extent in in-distribution data." Calibration methods smoothes the overconfident DNN outputs. So, I am not sure what you mean here. Am I missing something here?

Page 4-5: "Uncertainty estimation produces good calibrations in ID data, but often fails in the presence of dataset shift (Ovadia et al., 2019).  On the other hand, we expect dataset shift to manifest in an unusually large self-supervisionloss (Su et al., 2020).  By combining these two factors, we obtain a reliable detection signal."
Yes, uncertainty estimates tend to be poor if there is a data shift. But, it is not that they turn completely useless. In think, by combining the self-supervision loss and uncertainty, you compensate for the loss in uncertainty quality and recover from the decreased ability to detect uncertain/novel/OoD cases. I would rework these sentences. But, that is my interpretation. So, this might be a preference issue.

**Final Rating Justification:**

I thank the authors for addressing my comments and updating the draft accordingly.

I think the paper makes a decent contribution to medical image analysis by showing how self-supervision can be integrated into segmentation pipelines. Moreover, the results are promising despite some computational limitations, such as small batch size for contrastive learning. With larger minibatches or new tricks from the contrastive learning area, the proposed methodology and framework also has a good potential to be improved and adapted to different domains and tasks.

I would love to see this work at the conference. So, I am updating my rating from Weak Accept to Strong Accept.

**Justification Of The Preliminary Rating:**

The paper is well-written and addresses an important problem in medical image segmentation with decent results and implications for further developments in the field. Other imaging-based specialties of medicine can also benefit from the solution(s) proposed here. i would love to see this work at the conference.

**Paper Type:**

both

**Questions To Address In The Rebuttal:**

Please see my comments in the weakness section. These form are really entertaining...

**Special Issue:**

yes

---

> ### Author Response · Authors · 2021-03-16
> **Response to AnonReviewer1 (1/2)**
>
> We thank the reviewer for the insightful comments and for raising interesting new questions. We hope we have addressed all concerns in the comments below and in the revised manuscript.
>
> **Regarding the better performance of the models with contrastive learning compared to edge detection**
>
> We definitely agree with the reviewer that contrastive learning makes more sense semantically, and also results in better OOD detection results. However, our main objective in to present an OOD detection solution that works with a variety of self-supervised architectures, not models that use a specific proxy loss. That is why we evaluate the proposed method both with edge detection and contrastive learning. But if one were to train the model with the goal of OOD detection in mind, we agree that contrastive learning would be the preferable proxy task to use.
>
> **Regarding the implementation of the contrastive learning loss**
>
> We thank the reviewer for their comments on contrastive learning, which show ample knowledge in the area. We have adopted a very simple contrastive learning method, which we illustrate in Figure 2. Each dataloader item consists of two randomly selected images $x_i$ and $x_j$, as well as a transformed one $\overline{x_i}$. We use only $x_i$ and the corresponding $y_i$ to train the segmentation task. Our self-supervision loss maximizes the cosine similarity in the feature space between $x_i$ and $\overline{x_i}$ and minimizes the similarity between $x_i$ and $x_j$ (Equation 1). We have used the largest batch size that fits in our memory, namely 16. We could have improved the contrastive learning method, but our goal in this work is not to present a specific contrastive learning algorithm but to show how a contrastive learning loss can be leveraged for OOD detection.
>
> **Regarding the generalization for M&M Vendor A data**
>
> Indeed, models trained with *M&M Vendor A* data definitely show the best generalization. We thank the reviewer for raising this relevant point, which we had not previously analyzed. After manually examining the data we think one possible explanation could be that this dataset seems to contain more variability in terms of left and right ventricular blood pool ratio than the other two. Other suggestions made by the reviewer (e.g. technology and procedural differences) may also contribute to this performance gap. As we state in the revised manuscript, *M&M Vendor A* data was acquired with a *Siemens Avanto* scanner, while images in *M&M Vendor B* and *Sunnybrook* were acquired with *Philips Achieva* and *General Electric Signa* scanners, respectively. We do observe to some extent a better image quality and less incidence of artifacts in the *M&M Vendor A* data. See, for instance, the website for the *Multi-Centre, Multi-Vendor & Multi-Disease Cardiac Image Segmentation Challenge* https://www.ub.edu/mnms/ (image *(4) Siemens* belongs to *M&M Vendor A* in the released data, whereas *(3) Philips* belongs to *M&M Vendor B*). Please also see an example for a *Sunnybrook* scan at https://www.midasjournal.org/browse/publication/658.
>
> The resolution is, however, not a contributing factor, as all images have very similar resolutions and we further unify this during the pre-processing phase. We unfortunately have no information about the technical capacities of operators or procedural details other than those stated in the cited publications.
>
> **Regarding the use of 3-fold cross-validation**
>
> We definitely agree with the reviewer that a greater number of cross-validation folds would be preferable. We decided to perform 3-fold cross validation due to constraints in the GPU space we are able to allocate and the number of models overall that we needed to train for the evaluation. For the same reasons, we are not able to repeat the experiments for the rebuttal with a greater number of cross-validation folds. We believe that the relatively low standard deviations of the results show that these would not change significantly with a greater number of runs. However, we aim to increase the number of cross-validation runs in future works.

---

> ### Author Response · Authors · 2021-03-16
> **Response to AnonReviewer1 (2/2)**
>
> **Regarding the use of contrastive learning alongside edge detection**
>
> We thank the reviewer for this insightful comment. The reason we did not perform this experiment is that most self-supervised architectures found in the literature use only one proxy task. The main takeaway of the paper is that a self-supervised loss calculated at test time together with an uncertainty estimate is a reliable signal for OOD detection. While we evaluate our method with the proxy tasks of edge detection and contrastive learning, the method can augment self-supervised architectures that use any proxy task. We believe the flexibility of approaches for OOD detection is important, as in practice OOD detection is a secondary goal.
>
> However, it would certainly be possible to train a model with both proxy tasks explored in this work, and we do believe that they would complement each other, as both encode geometric information. Precisely because the learned information is similar, we do not believe that the OOD detection results would be significantly improved. But we would as well be interested in the outcome and will explore this in the future.
>
> **Regarding confusing statements and detailed suggestions**
>
> We would like to thank the reviewer for the detailed suggestions and for pointing out confusing statements in the text. We have refined the writing and believe the manuscript has improved compared to the original version.
> * **Paragraph in page 2 about requiring manual annotations:** We have rephrased this paragraph in the introduction (page 2).
> * **Figure 1 caption:** We realize this sentence was not well-written. What we meant to say is that the data that the model is trained with is OOD from the perspective of the ID data. We have modified the caption text, and now also mention the name of the dataset.
> * **Increasing softmax probabilities to a greater extent (page 3):** We have rephrased this sentence in the text. Besides utilizing temperature scaling, the ODIN method adds small perturbations to the model inputs. These perturbations increase the separation of maximum softmax values between ID and OOD data. By increasing the separation, OOD samples can be more easily detected.
> * **Improving uncertainty estimation (page 4-5):** We agree that this perspective better illustrates the motivation, and have adapted the specified sentences to reflect this.

---

### Official Review · AnonReviewer3 · 2021-03-07

**Confidence:** 4
**Preliminary Rating:** 2
**Final Rating:** 3

**Summary:**

The work presents a novel method for Out-of-distribution detection on CMR data for left ventricle segmentation. It proposes and evaluates two self-supervised proxy tasks for detection of these samples (contrastive learning and edge detection). The work is evaluated on publicly available data sets (sunnybrook and M&M callenge data) and shows certain improvements over current methods in literature.

**Strengths:**

The paper addresses a timely topic for identification of samples, which are out-of-distribution at test time. The approach is evaluated for a segmentation task and delivers promising results. Their method is compared to other state-of-the-art OOD methods.

**Weaknesses:**

1) As far as I understood (e.g. Fig. 2), the proxy task is learned together with the task of interest (segmentation). I am wondering how this effects the performance of the segmentation network G itself, in comparison to isolated training of the segmentation network. Does the performance degrade, when OOD is an additional goal?
2) It is known that segmentation networks for LV perform very well on center slices but have problems in the apex and base regions, because such regions look very different from the majority of slices.  Have you used 2D or 3D segmentation networks? Is the OOD detection based on 2D or 3D samples? Your OOD detection might be too sensitive towards these regions. Please add additional result descriptions showcasing the performance of OOD in center slices vs. apex vs. base.
3) From the description in sect. 4, the evaluation strategy using these 3 dat sets is not completely clear to me. It would be helpful to provide an illustration showing the crossvalidation and the respective ID and OOD settings.
4) In the $L_{ss}^C$, , segmentation loss is not mentioned.
5) The narrative of the paper is easy to follow, however, some statements are confusing and should be changed:
- "require no modification in the network architecture or training procedure" (p.2) -> There might be no changes necessary to a segmentation network itself, yes, but the presented approach might degrade the performance of the target task (see point 1). I think, changing the loss, such as in Eqn. 2, actually changes the training procedure.
-  "Self-supervised tasks do not require manual annotations, and so the performance in test samples can be assessed"(p.2) this statement is not clear to me
- "Unlike current state-of-the-art, the proposed approach does not require the use of a speciffic proxy task, or training the model with the explicit goal of OOD detection [...]across three CMR datasets and for two different proxy tasks." (p.2) First you argument that you do not need a proxy task and then you are actually using 2 proxy tasks. I guess you want to highlight that you do not require additional labels for the proxy tasks? Similar statements can be found at the end of the related work.
- Fig. 1: From which dataset is the ground truth segmentation?
- I recommend using Violin plots instead of boxplots for the results. Please indicate mean, and median in the violin plots.
6) What is the difference between M&M data sets and  sunnybrook? Does Sunnybrook data come from a different vendor than the two M&Ms, which would qualify it for such an OOD analysis?


**Deanonymize Review:**

no

**Detailed Comments:**

- Ejection Fraction, not Ejection Function (p2)

**Final Rating Justification:**

I thank the authors for futher clarification and for the willingness to make further changes to the document. I feel like that most of the points I raised have been addressed now. Therefore, I change my rating from weak reject to weak accept.


**Justification Of The Preliminary Rating:**

The paper addresses a relevant topic and the results show an improvement of their results over existing work. However, the authors should address the mentioned issues that have been raised. From their presentation, it is not clear to me whether their ODD method changes the performance of the target task, which would be undesirable.

**Paper Type:**

methodological development

**Questions To Address In The Rebuttal:**

point 1-3 and point 6

**Special Issue:**

no

---

> ### Author Response · Authors · 2021-03-16
> **Response to AnonReviewer3 (1/2)**
>
> We thank the reviewer for the thorough review and helpful suggestions. We agree that several of the questions raised are significant for assessing the performance of our method and were not answered clearly in the original manuscript. We have addressed all concerns in the revised manuscript, which we elaborate in the following.
>
> **Regarding the segmentation performance with and without self-supervision (point 1)**
>
> Based on the feedback, we have included results for the segmentation models without any proxy task in Appendix A, which is indeed relevant for assessing the results of the self-supervised networks. The segmentation performance of the same architecture without a proxy task is similar to the performance of the self-supervised models, slightly worse for the *Sunnybrook* dataset and slightly better than the edge-detection architecture for the other two datasets.
>
> As our focus lies in the utilization of the self-supervised loss for OOD detection rather than on finding an optimal proxy task, we did not perform experiments to optimize the weighting between the segmentation and self-supervised losses (we weighted these equally). One thing we noticed during training is that the self-supervised loss converges before the segmentation loss for both architectures. Particularly for the case of edge detection, this always seems to occur during the first 50 epochs.
>
> We want to emphasize that we do not propose using these exact proxy tasks. Instead, the main contribution of the paper lies in leveraging the value of the self-supervised loss for OOD detection for models that are already trained in a self-supervised manner.
>
> **Regarding the performance on apex, center and base regions (point 2)**
>
> We thank the reviewer for pointing out this practical problem of Cardiac CMR segmentation. Indeed, from a manual qualitative exploration of the predictions we can confirm that the segmentation performance on ID data is worse in the apex and base regions, as well as on the boundaries of the ROI. However, we cannot provide quantitative results, as each region does not occupy the same slice ranges for all data points, and we have no annotations specifying where each region starts and ends for each case.
>
> As far as the OOD detection goes, we use 2D segmentation networks, but calculate one  novelty score for each 3D volume by averaging the slice-wise (or voxel-wise) results. Whereas the self-supervised loss specifies one value per slice, uncertainty estimation methods provide one value per voxel. These are averaged to obtain a volume-wise score. We have added a paragraph at the end of the methodology section (page 5) to explain this more clearly. It is possible that the novelty values in the center and base regions contribute to a greater (or lesser) extent to the volume-wise score. This would be interesting to explore but is outside the scope of this work, since we would ideally require annotations specifying – for each subject – which slices are occupied by each region.
>
> **Regarding the evaluation strategy (point 3)**
>
> We want to thank the reviewer for their valuable feedback. Based on their suggestion, we included a graphical illustration in Appendix D to illustrate our evaluation strategy. Moreover, we have rephrased the corresponding paragraph in the manuscript (page 6). We now explain our evaluation setup in the following way: “In turn, we consider each of the three datasets as ID and the other two as OOD. We divide the ID cases into three folds to perform cross-validation. For each cross-validation run, we train a model with the *ID train data* made out of two folds and evaluate it with the third fold, which is the *ID test data*. For the OOD detection, we use one OOD dataset and the ID train data to select the best hyperparameters and evaluate the detection performance on the second OOD dataset and the ID test samples. We average the results of using each of the two OOD datasets for the evaluation, and report the mean and standard deviation of the three-fold cross-validation.”
>
> **Regarding the segmentation loss in Equation 1 (point 4)**
>
> Equations 1 and 2 specify only the self-supervised loss terms, not the entire loss. This is shown in Figure 2, where we refer to the segmentation loss as $\mathcal{L}_\text{seg}$. The reason that the edge-detection loss seems to include segmentation terms is that we treat edge detection as a segmentation task. The edge detection loss thus consists of Dice and Binary Cross Entropy terms between the edge detection prediction $\hat{h_i}$ and the edge detection mask $h_i$.

---

> ### Author Response · Authors · 2021-03-16
> **Response to AnonReviewer3 (2/2)**
>
> **Regarding detailed suggestions and confusing statements (point 5)**
>
> We would like to thank the reviewer for the detailed suggestions and for pointing out confusing statements in the text. We have refined the writing and figures accordingly.
> * **Changes in network architecture and use of a specific proxy task:** Our intention while writing this passage was to state that the calculation of the novelty score is not dependent on a specific proxy task. We agree that this was confusing in the original manuscript and explain our intention more clearly in the revised version. While we train the models presented in this work with either edge detection or contrastive learning proxy tasks for the sake of the evaluation, the novelty could be calculated in the same way for any other self-supervised task. Segmentation architectures increasingly use self-supervised proxy tasks for a number of reasons, and the proposed novelty score can augment these architectures with OOD detection. The main takeaways of the paper are the introduction of a novelty score that uses the test-time self-supervised loss together with uncertainty estimates, and the fact that this signal reliably detects OOD samples for two different self-supervised architectures. We have made this more clear throughout the text.
>
> * **Assessing the performance on test samples:** We have rephrased this sentence in the introduction (page 2) and hope that our meaning is better explained. We meant to say that – for self-supervised tasks – the loss between predictions and target values can be calculated on the test data, as the target values are derived from the input images. This is not the case for supervised tasks, which use manual annotations as target values.
>
> * **Precedence of the example in Figure 1:** The image, as well as the ground truth segmentation, are from the *M&M Vendor A* dataset. We thank the reviewer for pointing out that we had not made this explicit in the image or caption and have modified the image accordingly.
>
> * **Use of violin plots:** We thank the reviewer for this suggestion. We have replaced the boxplots with violin plots and now indicate both the mean and the median for each method. The violin plots also contain boxplots visualizing the first and third quantiles. We hope that this new representation provides a more intuitive understanding of the distribution of novelty values for ID and OOD data.
> * **Ejection Fraction:** We have corrected the use of this term and thank the reviewer for pointing this out.
>
> **Regarding the differences between datasets (point 6)**
>
> We thank the reviewer for raising this point. We realize that we left out important details about the datasets in the original manuscript and now specify the vendors and scanner models. The differences between the datasets lie both in the vendor and the acquisition center. Due to space constraints we are not able to go into detail in the revised manuscript, but have adapted the text in page 6 (top) to clarify this.
>
> The data released for the *Multi-Centre, Multi-Vendor & Multi-Disease Cardiac Image Segmentation Challenge* (2020) was specifically designed to evaluate the generalization ability of deep learning segmentation models. In total, data from four vendors was gathered. Data from two of these vendors was publicly released with ground truth annotations, and this forms the datasets that we refer to as *M&M Vendor A* and *M&M Vendor B*. The images were acquired with **Siemens Avanto** and **Philips Achieva** scanners, respectively. The two datasets also differ in terms of acquisition center. For more information about this data please see https://zenodo.org/record/3715890.
>
> The *Sunnybrook* data was released for the *MICCAI 2009 Cardiac MR left ventricle (LV) segmentation challenge*. This data was provided by the *Sunnybrook Health Sciences Centre* in Toronto. Images were acquired with a **General Electric Signa** scanner. For further details please see https://www.midasjournal.org/browse/publication/658.

---

### Meta-Review · Area_Chair1 · 2021-02-22

**Recommendation:** Accept (Poster)

**Metareview:**

The paper proposes a method to indicate when a test sample differs from those in the training distribution. Thus, by detecting such OOD test cases, the proposed method aims to raise a flag that the learned method cannot be used on such OOD test data.
As the reviewers have pointed out, this is an important limitation of current methods and needs addressing.

While, I am positive about the paper (after the rebuttal stage), some issues raised by the reviewers still remain. The methodology relies on existing methods combining voxel-level uncertainty estimation with the value of the self-supervision loss. Empirical analysis doesn't employ existing methods for outlier/novelty detection, for which a lot of literature exists (both without and with deep learning). The employed baselines are standard DNN-based methods. For the self-supervised task of edge detection, the method relies on edge locations that is a very laborious task.

**Paper Type:**

methodological development

---

### Decision · Program_Chairs · 2021-03-31

Accept